# The Effect of Stereoregularity and Adding Irganox 1076 on the Physical Aging Behavior of Poly(1-trimetylsilyl-1-propyne)

**DOI:** 10.3390/polym15092172

**Published:** 2023-05-03

**Authors:** Vladimir Makrushin, Anton Kossov, Viktoriya Polevaya, Ivan Levin, Denis Bezgin, Dariya Syrtsova, Samira Matson

**Affiliations:** A.V. Topchiev Institute of Petrochemical Synthesis, Russian Academy of Sciences, Leninsky Prospect 29, 119991 Moscow, Russiasyrtsova@ips.ac.ru (D.S.)

**Keywords:** poly(1-trimethylsilyl-1-propyne), high free volume polymer, membranes, gas permeation, aging

## Abstract

The effect of the stereoregularity of poly(1-trimethylsilyl-1-propyne) [PTMSP] (*cis*-content from 50 to 90%) on physical aging was investigated by measurement of the gas permeability. Films from pure PTMSP as well as those with the addition of the antioxidant Irganox 1076 were exposed to the air. The permeability of pure PTMSP films increases with an increase in *cis*-stereoregularity and correlates with an increase in interchain distances (according to X-ray analysis). For pure PTMSP films, the most significant aging (up to 50% of permeability drop) was observed for polymers with mixed microstructure, and the slowest aging (10–30% of permeability drop) was observed for polymers with *cis*-regular structure. For PTMSP films with added Irganox 1076, some decrease in permeability with time is also observed. The addition of Irganox 1076 to PTMSP in mixed as well as *cis*-enriched configurations visibly slows down aging. In the case of *cis*-regular PTMSP with a slow aging rate, the introduction of an antioxidant does not provide any advantages. The high stability of *cis*-regular PTMSP demonstrates the possibility of obtaining more stable membrane materials with the highest equilibrium state of the polymer selective layer prepared by casting solution.

## 1. Introduction

Si-containing poly(1-trimethylsilyl-1-propyne) [PTMSP] is one of the best-known representatives of the class of polymers of disubstituted acetylenes [1,2]. PTMSP has an extremely high free volume and microporous structure, which provide properties unusual for convenient glassy polymers, namely, very high permeability, which increases with increasing size of the penetrant molecule [3]. More interestingly, PTMSP demonstrates the significant increase in separation selectivity of the n-butane/methane mixture, caused by a strong decrease (by about 90%) in the methane permeability in the mixture compared to the individual permeability [4,5,6].

Glassy polymers, including PTMSP, demonstrate physical aging, i.e., a decrease in the polymer fractional free volume and a significant decrease in permeability over time. Physical aging is due to the non-equilibrium state characteristic of glassy polymers at temperatures below the glass transition temperature. The segmental mobility of macrochains leads to a decrease in the non-equilibrium excess of the free volume fraction. PTMSP has a very high free volume and, as a result, exhibits faster aging compared to other typical glassy polymers [7,8,9,10].

Generally speaking, the rate of physical aging is affected by a number of factors, such as free volume, temperature, polymer structure, and even polymer film thickness. The dependence of the rate of the structural relaxation process on the chemical structure of the polymer and related effects on the rigidity of the polymer chain and the fraction of free volume was shown in [11,12,13,14]. For example, the effect of polymer regularity on the aging rate was studied for atactic, isotactic, and syndiotactic polystyrene [12,15]. It was shown that the aging kinetics of polystyrene differ with different tacticities and revealed a correlation between the aging rate and chain stiffness. The authors hypothesized that an increase in chain stiffness slows down segmental mobility and reduces the rate of aging.

The specific structure of PTMSP is formed by a rigid polymer chain containing C=C bonds and bulky substituents. The main chain of PTMSP contains alternating double bonds, and units can be in the *cis*- and *trans*-configurations (Figure 1), and the ratio of *cis*-/*trans*-units in the macromolecule depends on the polymerization catalytic system [16,17,18,19]. In turn, the geometric structure of the macromolecule affects the supramolecular packing and properties of PTMSP [19,20,21]. Khotimsky and co-workers [19] studied the relationship between the molecular characteristics and morphology of PTMSP obtained on Nb and Ta pentachlorides and the properties of the polymer. It was shown that the gas permeability and solubility of PTMSP depend on the lengths of regular sequences of units with the same geometry and the associated effect on the equilibrium rigidity of macrochains.

Moreover, PTMSP obtained on Nb and Ta pentachloride-based catalysts also exhibits different physical aging. For one, PTMSP synthesized on the TaCl_5_ catalyst is subjected to a more rapid drop in permeability coefficients than PTMSP synthesized on the NbCl_5_ catalyst [7,20,22]. The authors supposed that the observed differences in the rate of permeability decline were related to the chain configuration, which is catalyst-dependent. An analysis of the kinetics of the decrease in the fractional free volume over time in PTMSP samples synthesized on two catalysts also showed that PTMSP synthesized on NbCl_5_ and containing more *cis*-units had a longer relaxation time than PTMSP synthesized on TaCl_5_ and containing more *trans*-units.

In this paper, we studied the effect of the degree of stereoregularity on gas permeability changes in time for PTMSP samples synthesized with traditional pentachloride-based catalysts, yielding polymers with mixed configuration, as well as with novel pentabromide-based catalysts, yielding *cis*-enriched polymers as high as *cis*-regular configuration. Furthermore, one of the ways to increase the stability of the PTMSP polymer matrix is to introduce additional bulk particles into the polymer matrix, which reduces the mobility of polymer chains [23,24,25,26]. Bearing this in mind, we also studied the stability of the gas transport properties in PTMSP films with varying degrees of stereoregularity with the addition of the antioxidant Irganox 1076. This antioxidant has the bulk structure of a hindered phenolic compound (Figure 2). Irganox 1076 has shown its high efficiency in improving the thermo-oxidative stability of PTMSP [27]. At the same time, when placed in a polymer matrix, bulky Irganox 1076 molecules can restrict macrochain densification caused by relaxation phenomena.

## 2. Materials and Methods

### 2.1. Materials

Monomer 1-trimethylsilyl-1-propyne was obtained by reacting methylacetylene from the methylacetylene-allene fraction with an alkyl magnesium halide, which was followed by treating the reaction mixture with trimethylchlorosilane [28].

Monomer (99.8%) and the solvents cyclohexane (≥99.8%, Fisher Scientific, Waltham, MA, USA) as well as toluene (99.97%, Fisher Scientific, Waltham, MA, USA) were distilled three times over calcium hydride in a high-purity argon atmosphere before polymerization. Catalysts NbCl_5_ (99.9%, Fluka Chemie AG, Buchs, Switzerland) and NbBr_5_ (99.9%, ABCR, Karlsruhe, Germany ), cocatalysts Ph_4_Sn (>98.0%, TCI Ltd., Tokyo, Japan ), *n*-Bu_4_Sn (~98%, Fluka Chemie AG, Buchs, Switzerland), and antioxidant Irganox 1076 (Octadecyl β—(3,5-di- tert -butyl-4-hydroxy phenyl)- propionate) (Sigma-Aldrich Co., St. Louis, MI, USA) were used as they are.

Ar with purity at least 99.9999%, O_2_, N_2_ with purity at least 99.9%, CH_4_ with purity at least 99.5%, He and H_2_ with purity at least 99.9%, and CO_2_ with purity at least 98.0% were purchased from Linde (Linde Gas Rus, Balashiha, Russia).

### 2.2. PTMSP Synthesis and Characterization

The PTMSP samples used in this study were cooked according to the procedure described elsewhere [19].

The intrinsic viscosity values of the polymers were measured in CCl_4_ at 25 °C in air at atmospheric pressure using an Ostwald–Ubbelohde viscometer.

The spectra of ^13^C NMR PTMSP solutions (in C_6_D_12_) were recorded on an Avance spectrometer (Bruker BioSpin GmbH, Ettlingen, Germany) in the single-pulse mode with a broadband proton decoupling during the FID signal and a reduced decoupling power during the relaxation delay (duration 3 s), which made it possible to keep the signal amplification by the Overhauser effect. The spectral capture width was 250 ppm. The accumulation time of the free induction decay signal was 12–18 h. The spectra were analyzed, including the decomposition of complex spectral lines, using the ACD/Labs program (Advanced Chemistry Development, Inc., Toronto, ON, Canada, version 12.0 for Microsoft Windows). The quantitative calculation of the ratio of *cis-* and *trans*-units from the doublet signals of the ^13^C NMR spectra is given in our previous works, for example, in [19].

### 2.3. Preparation of Polymer Films

The film membranes from pure PTMSP were prepared by casting polymer solution (1.5% wt.) in cyclohexane onto cellophane. The film membranes from PTMSP with stabilizer were prepared by the same procedure from a polymer solution in which Irganox 1076 (2% wt.) was added. After the evaporation of the solvent for 7 days in air at 20 °C, the films were removed from the cellophane and evacuated under vacuum for 48 h. The dry film membranes had thicknesses of 80–90 μm. Directly after evacuation, the films were measured for gas permeability. Then, for a month between measurements, the film membranes were stored in the air at room temperature.

### 2.4. Investigation of Films by X-ray Diffraction

The diffraction patterns of the polymer films were obtained on a Rigaku Rotaflex RU-200 X-ray diffractometer (Rigaku Co., Ltd., Tokyo, Japan) with a rotating copper anode (characteristic radiation wavelength 0.1542 nm). Flat films were fixed in 6 layers on an aluminum frame. The exposure was carried out in the “transmission” geometry in the angular range of 2.5–50 degrees in 2θ according to the Bragg–Brentano scheme. Then the resulting diffraction patterns were processed using the Fityk program; after subtracting the background line, they were presented as a sum of several Gaussian peaks. The angular position of these peaks was recalculated into the interplanar distances using the Bragg equation. The peak areas were used to calculate the specific intensities of the maxima (from the ratio of the total intensity of all observed peaks), and the sizes of their characteristic coherent scattering regions (CSRs) were estimated from their integral widths using the Scherrer equation.

### 2.5. Transport Properties Determination

#### 2.5.1. Gas Permeability Measurement Method

The permeability of O_2_, N_2_, CO_2_, CH_4_, He, and H_2_ was determined by the differential method with a gas chromatographic analysis using the Crystalux 4000M gas chromatograph. He and Ar were used as gas carriers [29]. The pressure drop across the membrane was 1 atm, and the membrane cell temperature was 22–23 °C. The installation scheme is shown in Figure 3.

The gas permeability coefficient *P* was found according to the equation:(1)$$P=\frac{l}{∆p}\cdot \frac{273\;K}{T\cdot A}\cdot \frac{p_{atm}}{76\;cm\;Hg}\cdot (\frac{∆V}{∆t})$$
where *l* is the film thickness, cm; ∆*p* is the pressure drop, cm Hg; *T* is the membrane cell temperature, K; *A* is the membrane area, cm^2^; *p_atm_* is the atmospheric pressure, cm Hg; is the permeate flow rate, cm^3^/s.

The ideal selectivity was calculated as follows:(2)αAB=PAPB
where *P_A_* и *P_B_* are permeability coefficients for gases *A* и *B*.

The experimental percentage error of the method was 5–7%.

#### 2.5.2. Gas Solubility Measurement Method

Gas sorption measurements were carried out using an original setup developed at the Membrane Gas Separation Laboratory of the Institute of Chemistry, Russian Academy of Sciences. The installation diagram is shown in Figure 4.

In the experiment, a metal tube 8 with an inner diameter of 2 mm and a length of 90 mm was used as a gas cell. The amount of gas in the volume of the cell without the studied polymer was determined to estimate the volume of the empty cell. To do this, the gas cell was filled with the gas under study, and the amount of gas was detected using a gas chromatograph Crystalux 4000M with a thermal conductivity detector 11(TCD). The temperature detector was at 150 °C. Then the polymer film was cut into strips, which were placed in a gas cell. The mass of the film in the cell was calculated from the difference between the masses of the empty and filled tubes. After that, the time of saturation of the sample with the studied gas was determined, which was accomplished by keeping a constant amount of gas in the sorption space. To do this, a series of experiments were carried out, after which desorption was carried out by the gas carrier into the TCD to determine the total amount of gas. Further, having achieved the equilibrium value of sorption in the entire volume of the studied material, a gas with a known partial pressure was passed through the cell with the polymer. After complete saturation of the sample, the amount of gas in the cell was detected using TCD, and then the test gas was desorbed by evacuating the cell. The pressure was 0.1–6 atm. Based on the obtained data, sorption isotherms were created, and the solubility coefficient was calculated from the slope of the initial section of the isotherm (0.1–2 atm). The gas solubility coefficient was calculated with the equation:*S* = Δ*c*/Δ*p*, (3)
where *c* is the gas concertation in the polymer, cm^3^(STP)/cm^3^, and *p* is the pressure, cm Hg.

The experimental percentage error of the method was up to 15%.

## 3. Results and Discussion

### 3.1. Polymer Properties

Based on our previous works [19,21], the polymerization conditions for synthesizing PTMSP samples with targeted *cis*-/*trans*-ratio compositions were determined. The three PTMSP samples selected for this work differ in *cis*-content: from a mixed configuration (50% *cis*-units) in PTMSP (synthesized with a traditional NbCl_5_ catalyst) to *cis*-enriched (80% *cis*-units) and *cis*-regular (90% *cis*-units) configurations (PTMSP synthesized with NbBr_5_-based catalysts). Synthesis conditions and characteristics of PTMSP samples are given in Table 1.

### 3.2. Effect of the PTMSP Microstructure on Gas Transport Properties

O_2_, N_2_, H_2_, He, CO_2_, and CH_4_ pure gas permeability as well as ideal selectivity through the film membranes from pure PTMSP with different *cis*-contents were studied, and the obtained results are shown in Table 2 and Table 3.

As can be seen from Table 2, an increase in the permeability coefficients with an increase in the content of cis-units is observed for almost all gases. The greatest difference is observed with an increase in cis-units from 80 to 90%. The permeability growth is accompanied by an expected slight decrease in ideal selectivity for all gas pairs (Table 3).

Differences in gas permeability indicate differences in the overall level and structure of the free volume of PTMSP with different contents of cis-units. As is known, the configurational composition of a polymer chain (the ratio of cis- and trans-configuration units) largely determines the packing of macrochains in a polymer. Changes in the ratio of cis-/trans-units, even small ones, lead to changes in the packing density of macrochains. The packing density, in turn, determines not only the overall level of the free volume but also the structure of the free volume, namely, the size of the interconnected free volume elements through which the molecules of penetrant gases are transported. Wide-angle X-ray diffraction was used to study the supramolecular organization of PTMSP films with different cis-unit contents. The X-ray diffraction patterns are shown in Figure 5.

All X-ray diffraction patterns of polymers show the main reflection with a half-width ∆_1/2_°~3.5°–3.8° and additional diffuse maxima. The presence of additional reflections, in addition to the most intense one, indicates an increased degree of ordering of PTMSP compared to typical amorphous polymers. The presented patterns show differences for PTMSP with different content of cis-units. Thus, for PTMSP with a cis-regular configuration (90% of cis-units), the intensity of peaks with an angular position of ~20 and ~28 deg becomes the most noticeable compared to PTMSP with a lower content of cis-units (50 and 80%). Moreover, in the cis-regular PTMSP, the X-ray diffraction pattern also clearly reflects peaks with an angular position of ~36 and ~44 deg. These data serve as an indication of the greater ordering of the structure when moving from a mixed geometry to a cis-rich composition, with the most noticeable difference being observed when moving to PTMSP with a 90% cis-content, i.e., a polymer with a high degree of cis-regularity. At the same time, in the PTMSP series, with an increase in the cis-content of 50–80–90%, there is a slight but systematic increase in the interchain distance corresponding to the main maximum, from 8.83 Å to 9.01 Å (Table 4), which may indicate a decrease in the packing density of PTMSP chains with an increase in cis-regularity.

It can be assumed that the high degree of stereoregularity of PTMSP-90c allows the macromolecule to adopt the most ordered and rigid helical conformation compared to the PTMSP-50c and PTMSP-80c polymers with a lower degree of stereoregularity. The rigidity of cis-regular PTMSP macromolecules with bulky substituents in a helical conformation prevents close packing, which can be evidenced by an increase in interchain distances with an increase in the degree of stereoregularity in the series PTMSP-50c—PTMSP-80c—PTMSP-90c. At the same time, the regular configuration of PTMSP-90c macromolecules apparently contributes to a more ordered arrangement of macromolecules compared to less regular PTMSP samples, which is reflected in the diffraction scattering patterns of the studied polymer samples with different degrees of stereoregularity.

### 3.3. Effect of Aging on the Gas Transport Properties of PTMSP Films

The study of the influence of the PTMSP’s stereoregularity on the stability of its gas transport characteristics over time was carried out for one month. The samples were stored in the air. The relative changes in the coefficients of permeability P/P_0_ for PTMSP film samples of different configurational compositions for the freshly prepared and aged samples are shown in Figure 6. It can be seen that the cis-regular PTMSP-90c loses only 10 to 30% of the initial permeability values. The permeability decreases to a greater extent for methane and nitrogen and to a lesser extent for highly permeable gases. The most significant (about 50%) decrease in permeability coefficients after a month is observed for the PTMSP-50c sample with mixed microstructure. We believe that the differences in the aging rate of PTMSPs with different cis-contents are associated with differences in the thermodynamic rigidity of PTMSP macrochains. The rigidity of the chain is determined both by the quantitative ratio of units of the cis-/trans-configuration and by the lengths of regular sequences of units of one configuration or another, i.e., macrochain regularity. In the case of the PTMSP samples studied in this work, from mixed configuration composition to cis-regular, it can be assumed that the noticeably slower aging rate of PTMSP-90c is associated with the regular structure and rigidity of the macromolecules of this sample, which limit segmental mobility and reduce the aging rate.

Diagrams in Figure 7, Figure 8, Figure 9, Figure 10, Figure 11 and Figure 12 show comparisons of changes in the coefficients of permeability and selectivity of gases for films from PTMSP of different microstructures, prepared without and with the addition of the antioxidant Irganox 1076. Compared to all pure PTMSP samples, the introduction of the antioxidant Irganox 1076 in the polymer matrix reduces the permeability of all gases. This is due to the fact that sizeable antioxidant molecules occupy part of the free space in the free volume elements, which leads to a decrease in the free volume fraction of the polymer matrix.

Some decrease in permeability with time is observed for PTMSP films with an antioxidant in a similar vein to pure PTMSP. However, the degree of change in permeability for polymers with different configuration compositions is different. In the case of PTMSP with a *cis*-content between 50% and 80%, the loss of permeability after a month for films with the addition of Irganox 1076 is much lower than for films without an antioxidant (Figure 7 and Figure 9). Thus, for *cis*-enriched PTMSP-80c, the relative change in permeability coefficients after 1 month was no more than 20%, while for films without antioxidants, the decrease in gas flow was 40%. The films with Irganox 1076 exhibited improved ideal selectivity, in particular for CO_2_/CH_4_ and O_2_/N_2_ gas pairs, compared to pure PTMSP films (Figure 8 and Figure 10). It is important to note that the introduction of an antioxidant during the preparation of films from PTMSP with 50% and 80% *cis*-units made it possible to maintain the permeability of aged films above the permeability of aged pure PTMSP films: by 25% for PTMSP-50c and 35% for PTMSP-80c, with an increase in the CO_2_/CH_4_ selectivity of 50% and 15% relative to pure polymer, respectively.

A different picture is observed for PTMSP with a *cis*-regular configuration. As can be seen from the diagram in Figure 11, the addition of Irganox 1076 also reduces the level of permeability compared to pure polymer by reducing the available free volume, as in the case with PTMSP with a less regular geometric structure (PTMSP-50c and PTMSP-80c). However, the increased gas transport stability of pure films from this *cis*-regular polymer leads to the fact that the permeability of films with Irganox 1076, even freshly prepared, is lower than the level of an aged pure polymer film.

For aged PTMSP films without and with the addition of Irganox 1076, we experimentally obtained the values of the solubility coefficients using sorption isotherms that were linear in the range of 0.1–5 atm and then calculated the diffusion coefficients using the *P = D·S* expression (Table 5). An analysis of the presented data shows that the introduction of an antioxidant into the polymer matrix has different impacts on aged PTMSP samples with various configurations. In the case of aged films from pure and stabilized PTMSP with 50 and 80% *cis*-units, the solubility coefficients are either the same (for PTMSP-80cA_aged) or somewhat lower (for PTMSP-50cA_aged). However, the gas diffusion coefficients of aged films with Irganox 1076 (PTMSP-50cA_aged and PTMSP-80cA_aged) exceed those for films from pure polymers (PTMSP-50c_aged and PTMSP-80c_aged). Such a ratio of diffusion coefficients provides increased permeability coefficients in aged, stabilized films relative to pure aged films. In addition, this indicates that the available free volume of the aged polymer matrix of pure irregular PTMSP turns out to be lower with time than for samples with Irganox 1076 [7,30]. Another situation is for *cis*-regular PTMSP with the highest transport stability. The solubility coefficients of both pure and stabilized aged films coincide within the experimental error, while the diffusion coefficients in pure PTMSP-90c_aged are perceptibly higher compared to stabilized films (PTMSP-90cA_aged). Due to this pure *cis*-regular, PTMSP-90c_aged demonstrates higher permeability coefficients compared with films with Irganox 1076 (PTMSP-90cA_aged).

The obtained results show that the introduction of bulky Irganox 1076 molecules is effective in reducing relaxation processes in polymers with an irregular structure, for which noticeable aging was observed. In the case of *cis*-regular PTMSP, whose regular structure provides it with a slow aging rate, the introduction of an antioxidant does not give any advantages.

## 4. Conclusions

The permeability of films prepared from PTMSP with different *cis*-contents was studied over a period of 1 month to establish the effect of polymer stereo regularity on physical aging. The decrease in gas permeability as well as the increase in ideal selectivity of all PTMSP films aged in air for 1 month testify to physical aging. However, aging behavior differs considerably depending on the *cis*-content in PTMSP. In the PTMSP series containing *cis*-units from 50 to 90%, a correlation was established between an increase in stereoregularity and a decrease in the aging rate. The *cis*-regular PTMSP showed a slight permeability decrease after 1 month of storage, while the mixed configuration PTMSP was subjected to the most substantial decrease in permeability. The slowest aging of *cis*-regular PTMSP may be due to the regular structure and rigidity of *cis*-regular macromolecules, which limit segmental mobility and reduce the rate of aging.

The addition of Irganox 1076 lowers gas permeability and slightly increases the ideal selectivity. This can be attributed to a decrease in fractional free volume as a result of bulky antioxidant molecules occupying free volume. The films with antioxidants also show signs of physical aging, although the magnitude of relative changes in permeability for polymers with different *cis*-contents varies. The addition of Irganox 1076 to PTMSP in mixed as well as *cis*-enriched configurations visibly slows down the aging, and the level of gas permeability of the aged films with antioxidants is higher than the permeability of the aged films from pure PTMSP. In the case of *cis*-regular PTMSP, which demonstrates increased gas transport stability over time, the gas permeability of the fresh film with a stabilizer is already lower than the level of permeability of an aged pure polymer film. It has been shown that the introduction of the antioxidant Irganox 1076 into the PTMSP polymer matrix does not significantly affect the gas solubility coefficients, regardless of the configurational composition of the polymer.

The results reported in this paper show the effectiveness of the antioxidant Irganox 1076 as a gas transport stabilizer in the case of PTMSP with lower stereoregularity. In the case of *cis*-regular PTMSP, the introduction of an antioxidant does not provide any advantages. The highest stability of *cis*-regular PTMSP relative to polymers of a less regular structure indicates the possibility of obtaining samples with the most equilibrium state of the polymer selective layer in the process of obtaining membranes by casting solution.

## Figures and Tables

**Figure 1 polymers-15-02172-f001:**
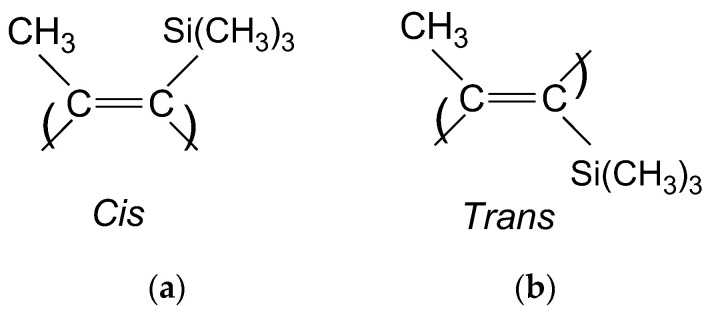
Models of the *cis*-unit (**a**) and *trans*-unit (**b**) of PTMSP.

**Figure 2 polymers-15-02172-f002:**
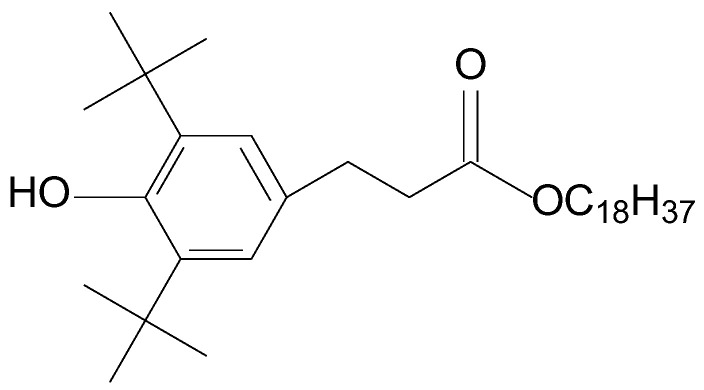
A structural formula for Irganox 1076.

**Figure 3 polymers-15-02172-f003:**
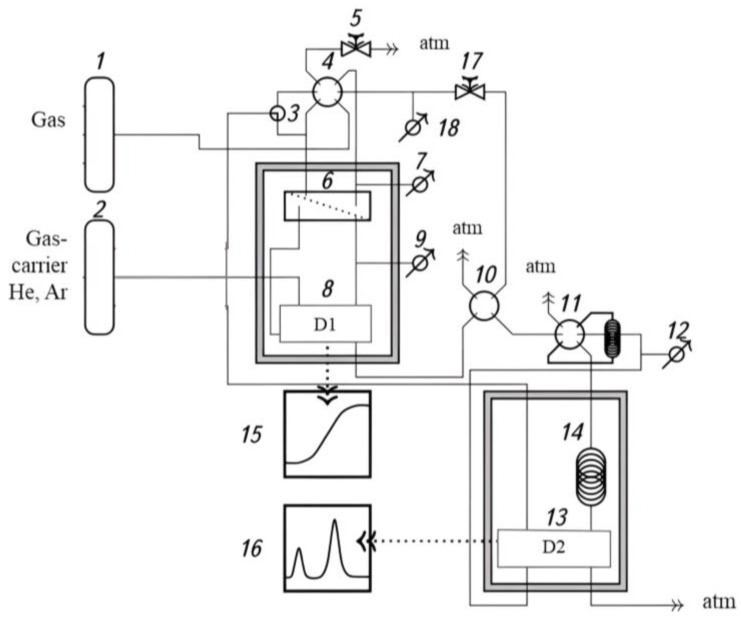
Schema of gas permeability set-up: 1 and 2—gas cylinders; 3—tee; 4, 5, 10, 11 and 17—valves; 6—membrane cell; 7, 9, 12, and 18—pressure sensors; 8 and 13—TCD, 14—gas chromatograph column; and 15 and 16—PC.

**Figure 4 polymers-15-02172-f004:**
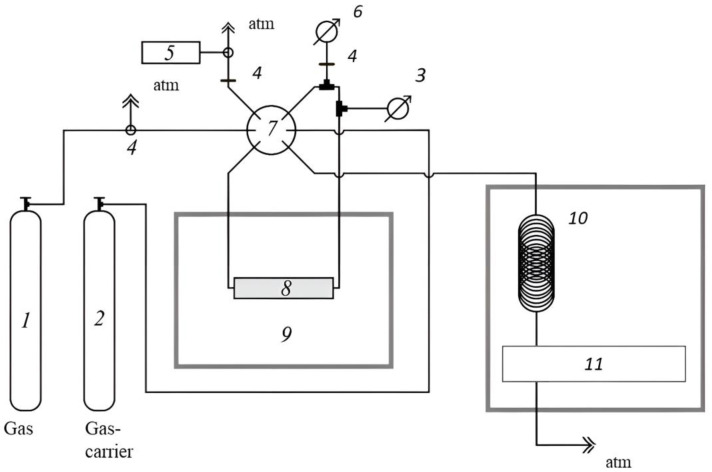
Schema of sorption set-up: 1 and 2—gas cylinders; 3—pressure sensor (≥1 atm); 4—valves; 5—vacuum cell; 6—pressure sensor (<1 atm); 7—valve; 8—tube with polymer; 9—thermostat; 10—GC column; and 11—TCD.

**Figure 5 polymers-15-02172-f005:**
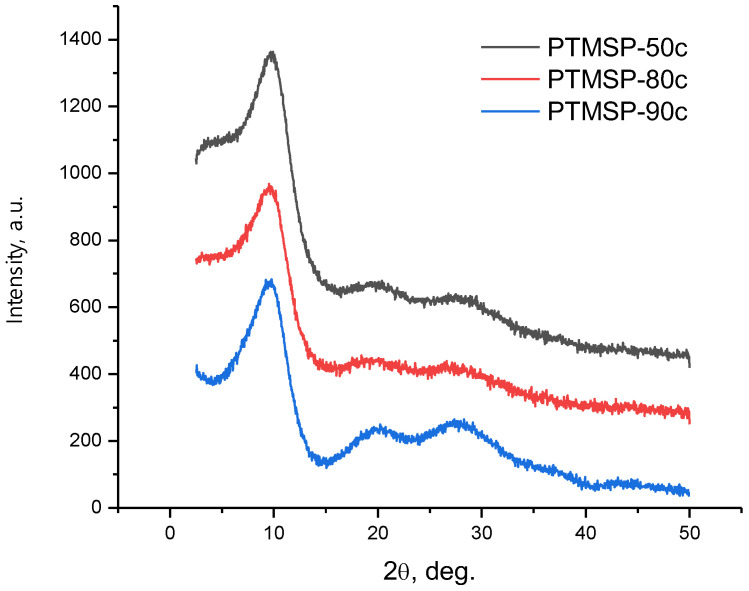
X-ray diffraction patterns of PTMSP with different *cis*-contents.

**Figure 6 polymers-15-02172-f006:**
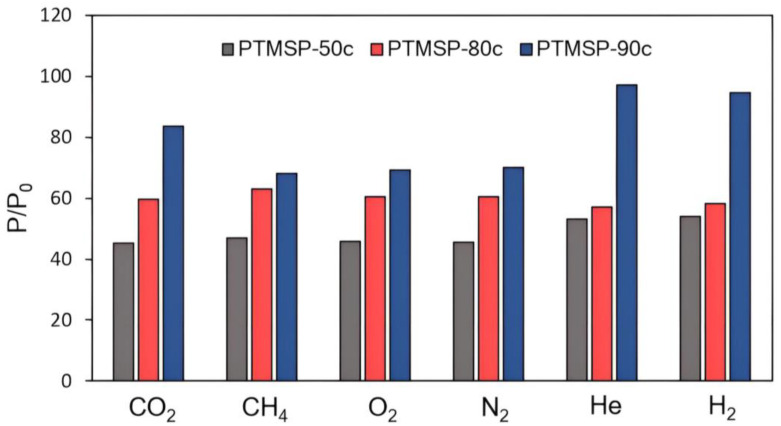
Relative changes in the coefficients of permeability P/P_0_ of films from PTMSP with different *cis*-contents during storage of samples in air for 1 month.

**Figure 7 polymers-15-02172-f007:**
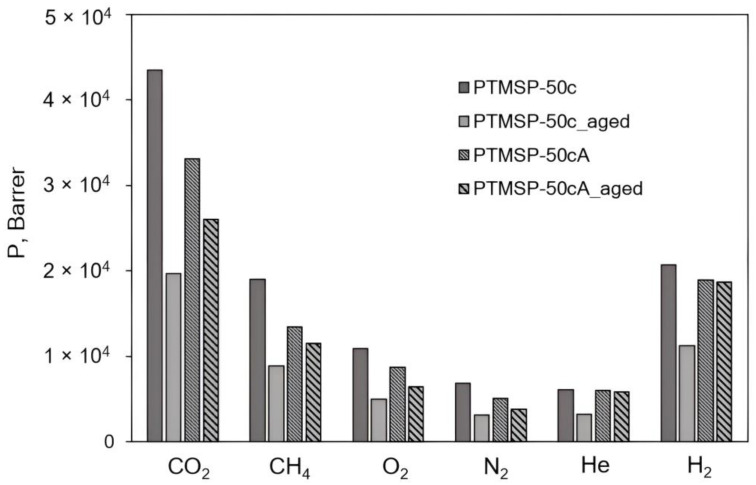
The effect of Irganox 1076 on the gas permeability of PTMSP films with the content of *cis*-units at 50%. The measurements were carried out for fresh films (PTMSP-50c film from pure polymer and PTMSP-50cA film with Irganox 1076) and for aged films stored in air for 1 month (PTMSP-50c_aged film from pure polymer and PTMSP-50cA_aged film with Irganox 1076).

**Figure 8 polymers-15-02172-f008:**
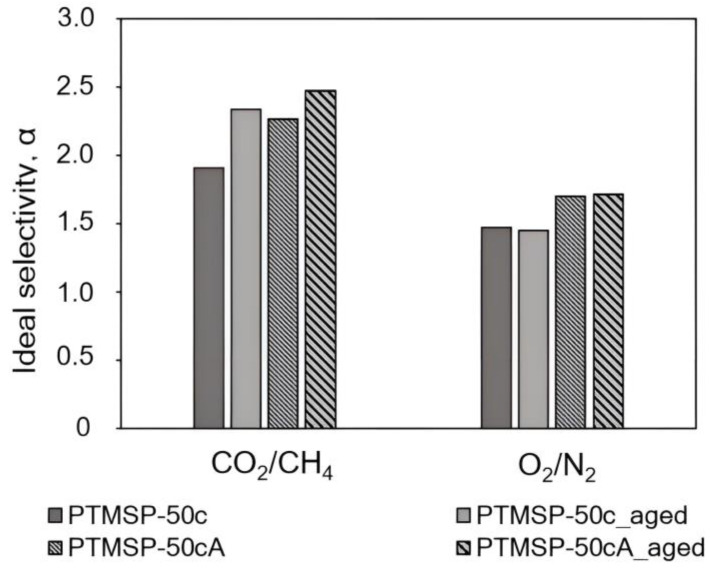
The effect of Irganox 1076 on the ideal selectivity of PTMSP films with the content of *cis*-units at 50%. The measurements were carried out for fresh films (PTMSP-50c film from pure polymer and PTMSP-50cA film with Irganox 1076) and for aged films stored in air for 1 month (PTMSP-50c_aged film from pure polymer and PTMSP-50cA_aged film with Irganox 1076).

**Figure 9 polymers-15-02172-f009:**
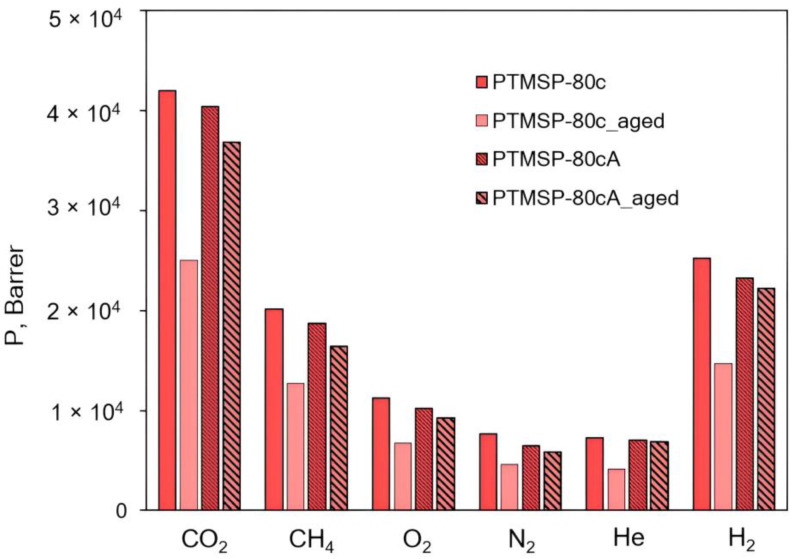
The effect of Irganox 1076 on the gas permeability of PTMSP films with the content of *cis*-units at 80%. The measurements were carried out for fresh films (PTMSP-80c film from pure polymer and PTMSP-80cA film with Irganox 1076) and for aged films stored in air for 1 month (PTMSP-80c_aged film from pure polymer and PTMSP-80cA_aged film with Irganox 1076).

**Figure 10 polymers-15-02172-f010:**
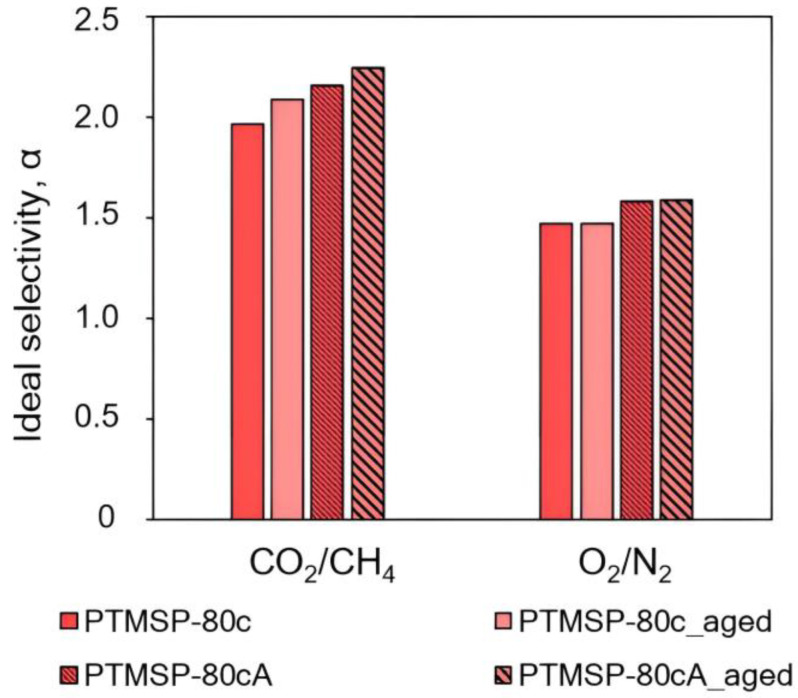
The effect of Irganox 1076 on the ideal selectivity of PTMSP films with the content of *cis*-units at 80%. The measurements were carried out for fresh films (PTMSP-80c film from pure polymer and PTMSP-80cA film with Irganox 1076) and for aged films stored in air for 1 month (PTMSP-80c_aged film from pure polymer and PTMSP-80cA_aged film with Irganox 1076).

**Figure 11 polymers-15-02172-f011:**
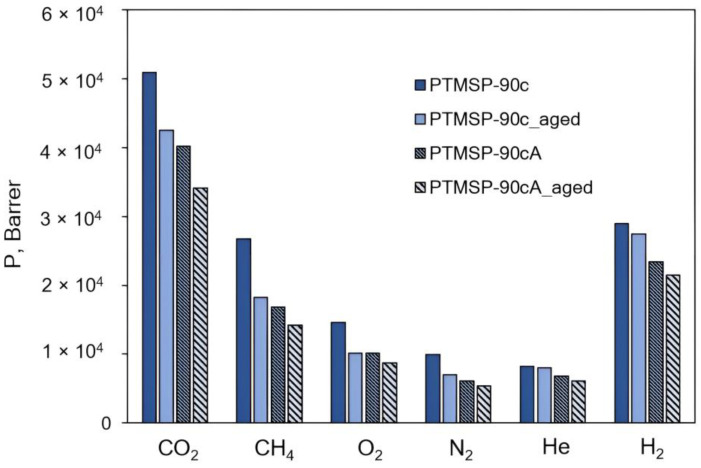
The effect of Irganox 1076 on the gas permeability of PTMSP films with the content of *cis*-units at 90%. The measurements were carried out for fresh films (PTMSP-90c film from pure polymer and PTMSP-90cA film with Irganox 1076) and for aged films stored in air for 1 month (PTMSP-90c_aged film from pure polymer and PTMSP-90cA_aged film with Irganox 1076).

**Figure 12 polymers-15-02172-f012:**
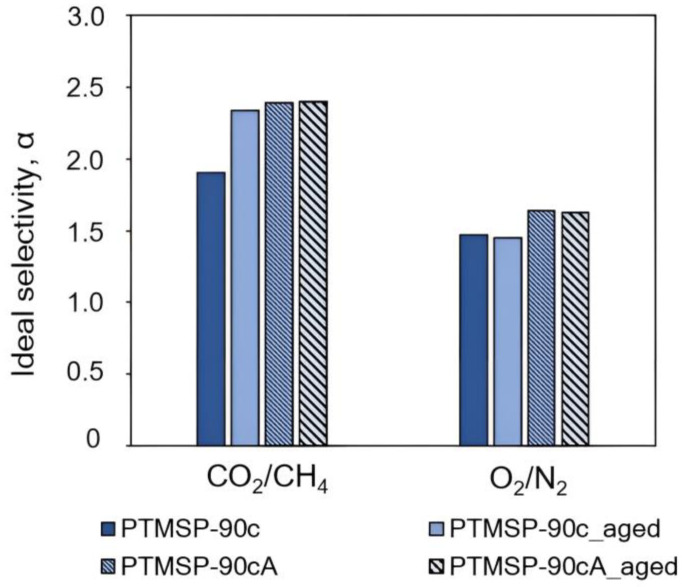
The effect of Irganox 1076 on the ideal selectivity of PTMSP films with the content of *cis*-units at 90%. The measurements were carried out for fresh films (PTMSP-90c film from pure polymer and PTMSP-90cA film with Irganox 1076) and for aged films stored in air for 1 month (PTMSP-90c_aged film from pure polymer and PTMSP-90cA_aged film with Irganox 1076).

**Table 1 polymers-15-02172-t001:** Synthesis conditions and characteristics of PTMSP.

Sample	Catalytic System ^1^	Yield, %	[η], dL/g	*Cis*-Content, %
PTMSP-50c	NbCl_5_	90	0.7	50
PTMSP-80c	NbBr_5_/Ph_4_Sn	70	2.7	80
PTMSP-90c	NbBr_5_/*n*-Bu_4_Sn	80	3.0	90

^1^ Polymerization conditions: [Mon]_0_ = 1 mol/L, [Mon]/[Cat] = 50, [Cat] = [Cocat] = 1; solvents: toluene for PTMSP-50c and cyclohexane for PTMSP-80c and PTMSP-90c; T = 25 °C and 24 h for PTMSP-50c; T = 80 °C and 72 h for PTMSP-80c and PTMSP-90c.

**Table 2 polymers-15-02172-t002:** Pure gas permeability of film membranes from PTMSP with different *cis*-contents.

Sample	Permeability Coefficient, P, Barrer *					
	CO_2_	CH_4_	O_2_	N_2_	He	H_2_
PTMSP-50c	43,500	19,000	10,900	6820	6100	20,700
PTMSP-80c	42,000	20,100	11,200	7630	7290	25,200
PTMSP-90c	50,900	26,700	14,600	9950	8230	29,000

* 1 Barrer = 1 × 10^−10^ [cm^3^ (STP) cm cm^−2^ s^−1^ cm Hg^−1^].

**Table 3 polymers-15-02172-t003:** Ideal selectivity of film membranes from PTMSP with different *cis*-contents.

Sample	Ideal Selectivity, α			
	CO_2_/CH_4_	CO_2_/N_2_	O_2_/N_2_	H_2_/CH_4_
PTMSP-50c	2.3	6.4	1.6	1.1
PTMSP-80c	2.1	5.5	1.5	1.3
PTMSP-90c	1.9	5.1	1.5	1.1

**Table 4 polymers-15-02172-t004:** Wide-angle X-ray scattering data for PTMSP with different *cis*-contents.

Sample	*Cis*-Content, %	Δ_1/2_ °	d, Å				
PTMSP-50c	50	3.5	8.83	4.43	3.17	---	---
PTMSP-80c	80	3.2	8.90	4.57	3.27	---	---
PTMSP-90c	90	3.4	9.01	4.53	3.21	2.46	2.03

**Table 5 polymers-15-02172-t005:** Gas permeability, diffusion, and solubility coefficients for aged films from pure PTMSP (PTMSP-50c_aged, PTMSP-80c_aged, and PTMSP-90c_aged) and Irganox 1076 (PTMSP-50cA_aged, PTMSP-80cA_aged, and PTMSP-90cA_aged) stored in air for 1 month.

Gas	P ^1^	D ^2^	S ^3^	P ^1^	D ^2^	S ^3^	P ^1^	D ^2^	S ^3^
	PTMSP-50c_aged			PTMSP-80c_aged			PTMSP-90c_aged		
CO_2_	19.7	1.6	127	25.0	2.6	97	43.0	4.6	92
CH_4_	8.9	1.9	47	12.7	4.2	30	18.1	4.7	39
O_2_	5.0	3.6	14	6.8	4.8	14	10.0	7.5	13
N_2_	3.1	2.9	11	4.6	4.0	12	7.2	6.3	11
He	3.2	-	-	4.2	-	-	8.0	-	-
H_2_	11.2	-	-	14.7	-	-	27.1	-	-
	PTMSP-50cA_aged			PTMSP-80cA_aged			PTMSP-90cA_aged		
CO_2_	26.0	2.8	93	36.8	4.2	95	34.0	3.5	98
CH_4_	11.5	3.3	35	16.4	4.0	31	14.2	3.4	42
O_2_	6.4	6.2	12	9.2	6.6	14	8.8	6.1	14
N_2_	3.8	3.7	8.7	5.8	13	12.7	5.4	4.5	11
He	5.8	-	-	6.9	-	-	6.2	-	-
H_2_	18.7	-	-	22.2	-	-	21.5	-	-

^1^ P·10^3^ [Barrer]; ^2^ D·10^5^ [cm^2^/s]; ^3^ S [cm^3^(STP)/cm^3^ cm Hg].

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
