# Peer review of "The Effect of Stereoregularity and Adding Irganox 1076 on the Physical Aging Behavior of Poly(1-trimetylsilyl-1-propyne)"

_polymers, 2023, doi:10.3390/polym15092172_

Round 1

Reviewer 1 Report (Previous Reviewer 2)

This work investigated the effect of stereoregularity and adding Irganox 1076 on the physical aging behavior of poly(1-trimethylsilyl-1-propyne). Although the quality of the manuscript has been improved, there is still a lack of proper interpretation of the result. Additionally, the limitations and inconsistent, unexpected trends obtained from this work have not been properly addressed in the revised version. The comparison of P, D, S for aged membranes with and without Irganox could mislead the reader. It would be better to include the P, D, S of the membrane before aging and it would be better to present some selected cases (not all gases). The graphs (especially Figures 7-12) are poorly presented without a major tick for the scale on the y-axis.  In addition, a clear definition of the technical unit like Barre must be provided. The ideal selectivity of the tested gases is very low for any real industrial applications.

Author Response

RESPONSE TO REVIEWER

Thank you for your perusal of our work and your comments. Our answers to your points are as follows.

Point 1: It would be better to include the P, D, S of the membrane before aging and it would be better to present some selected cases (not all gases).

Response 1: We agree that the data for P, D, S of freshly prepared  PTMSP films are of great interest. However, obtaining correct experimental data on the solubility coefficients for the freshly prepared samples had certain experimental difficulties due to the fact that the determination of the gas solubility coefficients was based on obtaining sorption dependences. Carrying out such studies requires considerable time in relation to obtaining data on gas permeability coefficients, and the most significant change in the structure of the matrix over time occurs just at the initial stage. Accordingly, sorption results obtained for gases studied at the very beginning may turn out to be irrelevant both to the values obtained later for other gases, and to the permeability coefficients of these gases obtained in the other period of time. This would, to a large extent, doubt the veracity of the diffusion coefficients calculated in this case. In the case of determining the permeability coefficients and permeability selectivity, we controlled the correctness of the experimental data for fresh films including by repeated experiments for the gases studied in the beginning of experimental series. Therefore, in order to avoid possible errors and discrepancies, we preferred to present the results obtained with high reliability for all the parameters presented.

Point 2: The graphs (especially Figures 7-12) are poorly presented without a major tick for the scale on the y-axis.  

Response 2: We have corrected the figures 7-12.

Point 3: In addition, a clear definition of the technical unit like Barrer must be provided.

Response 3: Barrer – it is a special unit that is traditionally used to estimate gas permeability coefficients in a wide range of values through polymer films. The Barrer dimension is listed below Table 2.

Point 4: The ideal selectivity of the tested gases is very low for any real industrial applications.

Response 4: It is known that since the 1990s, PTMSP has been considered as a promising membrane material for separation, for example, for separating air components or separating components of hydrocarbon mixtures [Yampolskii, Y., Pinnau, I., & Freeman, B. D. (Eds.). (2006). Materials science of membranes for gas and vapor separation]. In this work, we focused on solving the main problem of PTMSP as a membrane material, i.e., increasing the stability of PTMSP over time, since it is the rapid and significant decrease in the permeability of PTMSP over time that limits its practical application [Langsam, M., & Robeson, L. M. (1989). Substituted propyne polymers—part II. Effects of aging on the gas permeability properties of poly [1‐(trimethylsilyl) propyne] for gas separation membranes. Polymer Engineering & Science, 29(1), 44-54; Tanaka, A., Nitta, K. H., Maekawa, R., Masuda, T., & Higashimura, T. (1992). Effects of physical aging on viscoelastic and ultrasonic properties of poly [1-(trimethylsilyl)-1-propyne] films. Polymer journal, 24(11), 1173-1180]. Therefore, the development of approaches to improve the membrane behavior of a material based on PTMSP is a very important task.

We were able to demonstrate the success of two approaches to solving the problem of aging: through the changing of the polymer configuration (using new catalytic systems based on Nb and Ta pentabromides) and through the introduction of bulky molecules of phenolic-type stabilizing agents. The data presented in this work on the ideal selectivity α, including those for the O2/N2 gas pair, remain at the level of the previously studied PTMSP obtained on traditional systems based on Nb and Ta pentachlorides [Yampolskii, Y., Pinnau, I., & Freeman, B. D. (Eds.). (2006). Materials science of membranes for gas and vapor separation]. At the same time, we have shown in our our that the gas flow through the studied samples stored in air for a long time is significantly higher than the gas flows through the aged PTMSP described earlier in the literature [Langsam, M., & Robeson, L. M. (1989). Substituted propyne polymers—part II. Effects of aging on the gas permeability properties of poly [1‐(trimethylsilyl) propyne] for gas separation membranes. Polymer Engineering & Science, 29(1), 44-54; Yampol'Skii, Y. P., Shishatskii, S. M., Shantorovich, V. P., Antipov, E. M., Kuzmin, N. N., Rykov, S. V., V.L.Khodjaeva & Plate, N. A. (1993). Transport characteristics and other physicochemical properties of aged poly (1‐(trimethylsilyl)‐1‐propyne). Journal of applied polymer science, 48(11), 1935-1944].

The application of membranes based on PTMSP samples synthesized in this work was not discussed separately in this work. However, It should also be noted that glassy PTMSP is known to have a record high gas permeability (for example, permeability of freshly prepared films PO2 = 6000 – 11500 Barrer) [Masuda, T., Isobe, E., Higashimura, T., & Takada, K. (1983). Poly [1-(trimethylsilyl)-1-propyne]: a new high polymer synthesized with transition-metal catalysts and characterized by extremely high gas permeability. Journal of the American Chemical Society, 105(25), 7473-7474; Nagai, K., Masuda, T., Nakagawa, T., Freeman, B. D., & Pinnau, I. (2001). Poly [1-(trimethylsilyl)-1-propyne] and related polymers: synthesis, properties and functions. Progress in Polymer Science, 26(5), 721-798; Khotimsky, V. S., Tchirkova, M. V., Litvinova, E. G., Rebrov, A. I., & Bondarenko, G. N. (2003). Poly [1‐(trimethylgermyl)‐1‐propyne] and poly [1‐(trimethylsilyl)‐1‐propyne] with various geometries: their synthesis and properties. Journal of Polymer Science Part A: Polymer Chemistry, 41(14), 2133-2155]. Such parameters are unusual for typical glasses and far exceed the gas permeability of one of the most highly permeable rubbery PDMS. Selectivity α O2/N2 at the level of ~ 1.5 – 1.6 in combination with high time-stable fluxes ensures the location of the new PTMSP near the Robson boundary [Yampolskii, Y., Pinnau, I., & Freeman, B. D. (Eds.). (2006). Materials science of membranes for gas and vapor separation, p.98]. Taking into account the very high permeability of PTMSP, the achieved parameters of stability and selectivity may make it possible to consider PTMSP, synthesized on bromides, as a membrane material for nitrogen recovery as a retentate in compact membrane modules for various purposes.

Best regards,

Samira Matson and co-authors,

A.V.Topchiev Institute of Petrochemical Synthesis, RAS

Reviewer 2 Report (New Reviewer)

Aging phenomena is one of important consideration for the gas separation membrane, which is generally covered by a dense skin. In this study, the effect of stereoregularity of PTMSP on the physical ageing was investigated; the effect of antioxidant Irganox 1076 was investigated. This study has the significance to the gas separation membrane development. Especially, inducing antioxidant chemicals is an easy way to obtain the anti-ageing membranes.

1.       The ageing of polymer decreases the free volume, and then the permeability. It would be better to present the relevant parameter of membranes during the ageing time, e.g. free volume, Tg, d-spacing.

2.       It would be better to present the solubility coefficient and diffusion coefficient of the fresh film, which will be helpful in understanding the change in permeability. 

Author Response

RESPONSE TO REVIEWER

Thank you for your careful reading of our work and review of our paper. We have answered each of your points below.

Point 1: The ageing of polymer decreases the free volume, and then the permeability. It would be better to present the relevant parameter of membranes during the ageing time, e.g. free volume, Tg, d-spacing.

Response 1: We undoubtedly agree with this comment. We are going to present results of our study of changes in FFV as well as structure of FV during ageing in PTMSP in dependence of configurational regularity in our near special work.

Point 2: It would be better to present the solubility coefficient and diffusion coefficient of the fresh film, which will be helpful in understanding the change in permeability. 

Response 2: We agree that the data on the solubility and diffusion coefficients of freshly prepared  PTMSP films are of great interest. However, obtaining correct experimental data on the solubility coefficients for the freshly prepared samples had certain experimental difficulties due to the fact that the determination of the gas solubility coefficients was based on obtaining sorption dependences. Carrying out such studies requires considerable time in relation to obtaining data on gas permeability coefficients, and the most significant change in the structure of the matrix over time occurs just at the initial stage. Accordingly, sorption results obtained for gases studied at the very beginning may turn out to be irrelevant both to the values obtained later for other gases, and to the permeability coefficients of these gases obtained in the other period of time. This would, to a large extent, doubt the veracity of the diffusion coefficients calculated in this case. In the case of determining the permeability coefficients and permeability selectivity, we controlled the correctness of the experimental data for fresh films including by repeated experiments for the gases studied in the beginning of experimental series. Therefore, in order to avoid possible errors and discrepancies, we preferred to present the results obtained with high reliability for all the parameters presented.

Thank you for your kind remarks to our manuscript.

Sincerely,

Samira Matson and co-authors,

A.V.Topchiev Institute of Petrochemical Synthesis, RAS

Reviewer 3 Report (New Reviewer)

The manuscript is a well organized paper as of now, not sure if it is post-review. The only recommendation is to improve the resolution of Figure 1. The chemistry structure is not in high quality. 

Author Response

RESPONSE TO REVIEWER

Thank you for your perusal of our work and your comments. Our answer to your point is as follows.

Point 1: The only recommendation is to improve the resolution of Figure 1. The chemistry structure is not in high quality. 

Response 1: We improved the quality of Figure 1.

Sincerely,

Samira Matson and co-authors,

A.V.Topchiev Institute of Petrochemical Synthesis, RAS

This manuscript is a resubmission of an earlier submission. The following is a list of the peer review reports and author responses from that submission.

Round 1

Reviewer 1 Report

The manuscript “The effect of stereoregularity and adding Irganox 1076 on the physical aging behavior of poly(1-trimethylsilyl-1-propyne)” studied the detailed change of gas transport properties of PTMSP and the mixed matrix membranes, based on the effect of the polymer chain regularity. In the reviewer’s opinion, the manuscript could be accepted after some revision, as followed:

1) The language of the manuscript should be improved, for example, there are many long sentences in this manuscript which are hard to understand, the authors should pay attention to the smoothness of the manuscript;

2) The data figures should be provided in the same style;

3) It seems that the gas permeability of the membranes were tested by the use of pure gases, this is in paradox with the description of Figure 2, please explain;

4) There is NO gas sorption data in the manuscript.

Reviewer 2 Report

This work investigated the effect of stereoregularity of PTMSP and the effect of Irganox 1076 (antioxidant) on the physical aging of the membrane. Though the research topic is interesting and has the potential to draw attention from readers in the membrane separation field, the manuscript still has too many flaws, especially in the experimental method and the discussion part. This manuscript requires a major revision. The major concerns are as follows.

1)     Experimental part

o   Please provide all equipment models and manufacturers for all characterization techniques

o   The chracterization condition for Wide-angle X-ray diffraction was missing.

o   The details for membrane preparation (polymer films) should be provided. In addition, the film thickness for all membranes should also be mentioned since the film thickness has a significant impact on the physical aging rate.

o   The condition history of the membranes such as how they were stored, treated, or used before the test could contribute to the gas performance behavior. I am wondering if one membrane sample is used for one gas in the permeability test. Please clarify.

o   Please provide the chemical structure of Irganox 1076.

2)     Result and discussion

o   In section 3.3, the authors state “It can be seen that the cis-regular PTMSP-90c loses on 10 to 30% of the initial permeability values. The permeability decreases to a greater extent for methane and nitrogen, and to a lesser extent for light gases”.  Please clarify the term “light gases”. Compared to He and H2, CO2 is not light but the permeability declined less than CH4, O2, and N2.

o   From Table 4, the estimated interchain distance of PTMSP-80c and PTMSP-90c seem to be close to each other, why the gas permeance and the physical aging of the two membranes are so different?

o   Please explain whether or not the addition of Irganox 1076 affect the affinity between the permeate gas and the membrane.

o   Comparing Figures 6, 8, and 10, why did the addition of Irganox 1076 result in less gas permeability reduction in the case of PTMSP-80c while for PTMSP-50c and PTMSP-90c gas permeability reduced more significantly once the Irganox 1076 was added.

o   The unit for gas diffusion coefficient (D) provided in Table 5 is wrong

o   Please show how the D is obtained or estimated.

o   The authors mentioned on page 12 that “the introduction of an antioxidant into the polymer matrix of all three PTMSP samples does not significantly affect the gas solubility coefficients.”  Such a statement is wrong for the case of PTMSP-50c in which the S of the membrane for CO2 changed from 127 to 93 and for CH4 changed from 47 to 35 after the addition of Irganox 1076.